applied mathematics/evolution

evolutionary game theory, adaptive dynamics, ecology and evolution

**Author for correspondence:**
Gregory J. Kimmel
e-mail: gregory.kimmel@moffitt.org

†These authors contributed equally.

# Two-dimensional adaptive dynamics of evolutionary public goods games: finite-size effects on fixation probability and branching time

Brian Johnson, Philipp M. Altrock† and Gregory J. Kimmel†

Department of Integrated Mathematical Oncology, H. Lee Moffit Cancer Center and Research Institute, Tampa, FL 33612, USA

PMA, 0000-0001-7731-3345; GJK, 0000-0001-9766-5399

Public goods games (PGGs) describe situations in which individuals contribute to a good at a private cost, but others can free-ride by receiving a share of the public benefit at no cost. The game occurs within local neighbourhoods, which are subsets of the whole population. Free-riding and maximal production are two extremes of a continuous spectrum of traits. We study the adaptive dynamics of production and neighbourhood size. We allow the public good production and the neighbourhood size to coevolve and observe evolutionary branching. We explain how an initially monomorphic population undergoes evolutionary branching in two dimensions to become a dimorphic population characterized by extremes of the spectrum of trait values. We find that population size plays a crucial role in determining the final state of the population. Small populations may not branch or may be subject to extinction of a subpopulation after branching. In small populations, stochastic effects become important and we calculate the probability of subpopulation extinction. Our work elucidates the evolutionary origins of heterogeneity in local PGGs among individuals of two traits (production and neighbourhood size), and the effects of stochasticity in two-dimensional trait space, where novel effects emerge.

# 1. Introduction

Emergence, evolution and persistence of subpopulations with differing traits or strategies remains an open topic for many systems in ecology and evolution [1–3], including cancer cell population dynamics [4]. Speciation events in evolution are often difficult to observe because they occur over long timescales [5,6]. By contrast, somatic evolution occurs over shorter timescales, but has also proven to be difficult to understand due to complications arising from limited sampling ability [7–10]. Evolutionary game theory can be used to explore and explain the emergence and disappearance of traits or strategies as the environment and the population composition change over time [11]. Evolutionary outcomes in population games can be studied using adaptive dynamics, which permits the study evolutionary branching [12] and coexistence in continuous trait or strategy space over time [13–15]. For example, this framework has been used to study branching of complex traits, such as animal migration [16], polymorphism in cross-feeding [17], and the emergence of cooperators and defectors in social dilemma games in more general terms [18].

A social dilemma among groups of individuals emerges when individuals in a group contribute to a common good that is shared independently of contribution. All individuals can tap into a benefit or common resource, e.g. via consumption. As the resource is depleted by a momentary evolutionary advantage of non-contributors which only consume, the benefit to each individual erodes, until there is very little to none left. In this example, one might call those who maximize their consumption (minimize their contribution) of resource 'defectors' or 'free-riders', due to the detrimental effect their actions have on others. Producers or 'cooperators' would be those who maximize their contribution. Previous work demonstrates that cooperative behaviour, which may seem counterintuitive to maximizing individual fitness, can be an evolutionarily feasible outcome in various games, such as those which introduce a reputation reward or those with nonlinear payoff functions [18–20].

Social dilemma situations have been found to describe aspects of interactions among cancer cells [4,21], banded mongooses [1], prairie dogs [22], predator inspecting fish [23], or in mobbing of hawks by crows [24]. Microbes provide public goods (PGs) by secreting useful chemicals [25,26]. Yeast cells synthesize and spill essential nutrients into their surroundings [2], and were observed to coevolve accordingly [27]. Such potentially complex interactions can strongly influence the eco-evolutionary dynamics between PG producers and free-riders. However, it has been less examined how these two distinct strategies emerge in the first place, which can be addressed by studying the adaptive dynamics, or evolutionary invasion analysis, of these population games [13,28].

A key assumption in models of PG dynamics is the effect of the PG on the payoff or fitness of an individual [29]. There can be diminishing returns that naturally lead to nonlinear benefit functions [30]. Examples of benefit functions include: linear, convex, concave and sigmoidal benefit as a function of the available PG [4,29,31–38]. Here, we consider a sigmoidal relationship between the level of PG and the resulting benefit, as well as a nonlinear cost of production. We are interested in the evolution of two key traits that change these relationships. First, we consider the size of the group among which the PGs game is played, also called the neighbourhood. Second, we consider the level of PG production of each individual. We are interested in the evolution of the population in this two-dimensional trait space. Intuitively, it might be clear that producers who exhibit high levels of production favour small neighbourhoods, whereas defectors with low levels of production can only survive if their neighbourhood sizes are sufficiently large. It has been unclear so far how these distinct strategies can emerge from initially homogeneous populations, and what role small population sizes play in the outcome of the evolutionary game.

We employ adaptive dynamics [13] to show how an initially monomorphic population in trait space can evolve into two distinct subpopulations comprised of producers and defectors, each with a different neighbourhood size. When the population size is large, the trait evolution is deterministic and easier to predict. However, when the population size is small, stochastic fluctuations become relevant. Our work elucidates the impact of stochastic effects and the shortcomings of the deterministic approach in the prediction of branching in two dimensions. We seek to understand how small populations behave and how the results deviate from those predicted by the deterministic theory. In recent years, the impact of small populations has been studied most notably in the work of Wakano & Iwasa [39], Claessen et al. [40], and Débarre & Otto [41]. These works have revealed that branching in stochastic systems is either delayed or does not occur, and extinction of certain types may occur following branching in one-dimensional trait space.

Our work extends the study of stochasticity to two dimensions. Specifically, in contrast to analogous one-dimensional trait space population games, studied in the works of Wakano & Iwasa [39], Claessen *et al.* [40], and Débarre & Otto [41], we observe that branching may be deterministically favoured only for a finite amount of time in our two-dimensional model: the monomorphic population can drift away from the region where branching is deterministically favoured, a feature not seen in the single trait games. Additionally, we expand the current understanding of extinction events, determining the cause of extinction and predicting extinction times of a given subpopulation. Because many relevant processes begin with small populations, further exploration of coevolving traits in finite populations is warranted.

This article is structured as follows. In §2, we introduce the evolutionary population game, how it proceeds over time, and the methods we use to analyse the system. In §3, we present our simulation results and compare them with analytical approaches that predict trait evolution. In §4, we summarize our findings, put them into context, and highlight interesting areas for future work.

# 2. Methods

## 2.1. Model introduction

Benefits that are produced and shared by a focal individual are often only available within a finite subset, or neighbourhood [42,43]. Furthermore, some individuals may interact more locally than others, as observed in the variation of group size within the same animal species [44]. In cells, chemical signals can alter the speed and direction of cell motility, influencing the size of the neighbourhood with which the cell interacts [45,46]. Regulation of neighbourhood size via motility or other yet unknown mechanisms in tumour cells may promote metastasis [47,48]. Additionally, neighbourhood or group size has been shown to be an important parameter in previous public goods games (PGGs), influencing the overall outcome of evolution in various games [36,49,50]. Therefore, we are interested in how neighbourhood size and production evolve when they become part of the selection process. It should be noted that this game applies to non-human interactions, as human behaviour is too complex to be characterized by the assumptions we use in our model.

Models of evolutionary PGGs often assume a linear relationship between benefits and the number of producers [51,52]. Yet the collective benefit of a PG may be nonlinear [31,36,53–55]. There may be increasing returns when PG increases from low to higher levels, and diminishing returns when levels further increase from high to very high levels (i.e. saturation), leading to a sigmoidal PG to payoff relationship. Nonlinear functions with continuous traits have been shown to be capable of branching with a single trait [18,56]. In Zhang *et al.*, the exploration of continuous investment versus probabilistic binary investment in finite populations shows that only continuous investment allows for branching [56]. We further explore continuous investment as it relates to scenarios in which there are two coevolving traits.

The nonlinear benefit function of our model was previously introduced in Kimmel *et al.* [36]. We consider a sigmoidal benefit proportional to the amount of PG shared among a fixed neighbourhood of individuals. While there has been less study of the shape of cost functions in the literature, we use a sigmoidal cost as well. To motivate this choice, we assume some low baseline level of production with a similarly low cost. Then, activating different machinery for further production is costly. However, ensuing increases in production are less costly, once the alternative machinery is activated [57]. One such example of two different types of 'machinery' can be found in cells which have multiple pathways for secreting cytokines [58]. A sigmoidal cost function models this behaviour while keeping a smooth structure, simplifying the analysis. Neighbourhood size, $n$, determines the subset of individuals among which the focal individual shares the PG. The population size, $N$, is fixed throughout the process, and we consider evolution in steps of generations between which the entire population is replaced by their offspring population, with change due to selection and random mutation effects.

To study adaptive dynamics, we model PG production and neighbourhood size as two continuous traits. In this way, the neighbourhood size could be a function of motility, a behaviour or trait which can vary in response to epigenetic or environmental changes [47]. The population game works as follows. Consider a focal individual with production level $y$ and a neighbourhood of size $n_y$ (of which the individual itself is a part). Then, by any probabilistic assembly, there are $n_y - 1$ neighbours, with their respective levels of production labelled $x_i$ ($i = 1, \ldots, n_y - 1$). The PG production traits, $y$ and $x_i$, are continuous variables between 0 and 1. In the evolutionary process, the neighbourhood size trait is

**Table 1.** The table shows the model parameters and the typical values used for simulations. Some values were taken from previous literature (as referenced); others were chosen deliberately for proof-of-principle. PG: public good. NS: neighborhood size.

| parameter | meaning | typical value | references |
|---|---|---|---|
| $\sigma$ | PG-independent benefit | 2.0 | [36] |
| $\beta$ | PG-dependent benefit | 5.0 | [36] |
| $\kappa$ | cost magnitude | 0.5 | [36] |
| $\omega$ | cost shape | 2.0 | this work |
| $\mu_x$ | probability of mutation in individual production | 0.01 | [18] |
| $\mu_n$ | probability of mutation in individual NS | 0.01 | this work |
| $s_x$ | production mutation standard deviation | 0.005 | [18] |
| $s_n$ | neighbourhood size mutation standard deviation | 0.2 | this work |

also a continuous variable between 1 and $N$. It should be noted that while the neighbourhood size trait is continuous, the actual number of neighbours determined at each generation is rounded to the nearest integer.

An individual's neighbourhood size determines the size of the pot from which it draws. However, the individual must share the good in the pot evenly with all of its neighbours. In our model, an important feature of the PG is that it is not depleted by an individual which benefits from its availability; it can be used multiple times. Therefore, the same PG can provide benefit to multiple individuals. Then, if a producer is in its own neighbourhood, the concentration will be maximized in the area immediately surrounding (which we still define as its 'neighbourhood' even though it is the only neighbour). However, the neighbourhood of a defector may contain producers, and the defectors can still benefit from the producers within their larger neighbourhood. If we consider the individuals as cells, then diffusion of the PG and movement of the defector cells would allow the defector cells to use the PG produced by the producers, even after the producers have already benefited from the same PG. Therefore, the model allows for the producers to produce only for themselves while still generating PG that is available to the defectors.

When the payoff is calculated, the neighbourhood size of the focal individual, $n_y$, is rounded to the nearest discrete integer between 1 (private good) and the population size, $N$. For each focal individual in each generation, $n_y - 1$ neighbours are drawn from a hypergeometric sampling of the population, excluding the focal individual. For further details, see electronic supplementary material, appendix A. We can then calculate the focal individual's available PG, $G$, as the arithmetic mean of its and its neighbour's production

$$G(y, n_y, x_1, \ldots, x_{n-1}) = \frac{1}{n}\left(y + \sum_{i=1}^{n-1} x_i\right). \tag{2.1}$$

If we allow **x** to denote the production values of the chosen neighbours, $x_i$ ($i = 1, \ldots, n_y - 1$), then the nonlinear (sigmoidal) benefit to the focal individual is [36]

$$B(y, n_y, \mathbf{x}) = \frac{1 + e^{\sigma}}{1 + e^{\sigma - \beta G(y, n_y, \mathbf{x})}}. \tag{2.2}$$

The benefit function can be altered by adjusting $\beta$, which controls for the PG-dependent benefit, and $\sigma$, which controls for the PG-independent benefit. Unless otherwise specified, we set $\sigma = 2$ and $\beta = 5$ (table 1), as we have shown that these parameters should, in principle, allow for coexistence [36] to be maintained.

Cost of PG production, $C(y)$, is determined only by the production of the individual. It is controlled by $\kappa$, which specifies its relative magnitude, and $\omega$, which specifies the shape

$$C(y) = \kappa \tanh\left(\frac{y}{1 - y}\right)^{\omega}. \tag{2.3}$$

The cost function shape parameter, $\omega = 2.0$, was chosen to replicate the general sigmoidal shape of the benefit function, saturating at high production, but also increasing slowly for low production, with

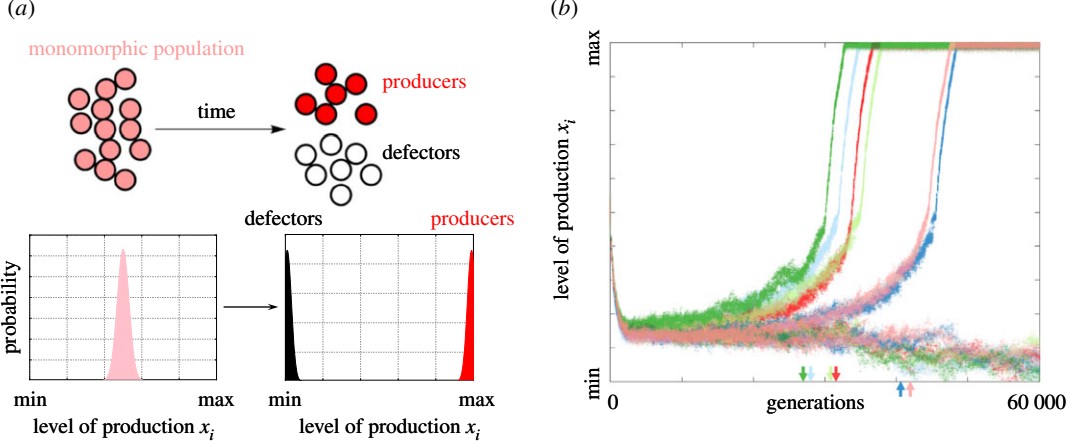

**Figure 1.** Macroscopic dynamics overview. (*a*) A schematic view of an initially monomorphic population branching into two distinct subpopulations, with colour indicating trait. Trait distributions are also shown for both the monomorphic and branched populations. (*b*) An example of the stochastic branching process, showing production over time for simulations with a population size of 1000. Each point represents an individual with its *y* value denoting its production and its *x* value denoting the generation. Six simulations with identical initial conditions are shown, with colour differentiating simulations. As predicted by deterministic theory, all of the simulations branch into two groups, one with high production and one with low production. The generation at which a simulation has branched is indicated by the arrow of the corresponding colour. The method for determining when branching occurs is detailed in §2.2. Time to branching varies even for this relatively large population size of 1000, highlighting the importance of stochasticity in our game. This stochasticity will be the focus of the following analysis, as shown in greater depth in figures 2–4. These simulations all use default parameters: $\beta = 5$, $\sigma = 2$, $\kappa = 0.5$, $\omega = 2$ (table 1).

the largest slope for medium production. The cost function magnitude, $\kappa = 0.5$, was chosen to match the binary, fixed cost in the model of Kimmel *et al.* [36].

Using benefit and cost functions, we can determine the payoff to the focal individual characterized by production level $y$ and neighbourhood size $n_y$, with neighbour production values **x**

$$P(y, n_y, \mathbf{x}) = B(y, n_y, \mathbf{x}) - C(y). \tag{2.4}$$

The initial traits of the population are seeded in production and neighbourhood trait space via two independent, normal distributions. Details of these distributions can be found in electronic supplementary material, appendix A. This monomorphic population then evolves over time in trait space. In certain scenarios, the population can branch and lead to two distinct subpopulations, as shown in figure 1*a*.

In our model, trait evolution follows the process proposed by Doebeli *et al.* [18]. Time advances in discrete generational steps during which each individual's payoff is calculated as previously stated. After a focal individual interacts with neighbours to determine its payoff, the payoff is then compared with the payoff of a second, randomly selected, individual, which we call the opponent. The opponent has some probability to replace the focal individual. Replacement probability is normalized by $\alpha$, which is equal to the difference between that generation's most and least fit individuals

$$\alpha = P_{\max} - P_{\min}. \tag{2.5}$$

Then, with probability $q$, the opponent replaces the individual and becomes the parent of an offspring in the next generation

$$q = \max\left(\frac{P_{\mathrm{opp}} - P_{\mathrm{focal}}}{\alpha}, 0\right). \tag{2.6}$$

If replacement does not occur, the focal individual is the parent. The offspring will have the traits of the parent, possibly altered by mutation.

The mutation process involves the two probabilities of mutation, $\mu_x$ and $\mu_n$, (table 1), such that on average, one out of every $1/\mu_x$ individuals is expected to mutate in a given generation for production, and one of every $1/\mu_n$ is expected to mutate in neighbourhood size. Mutation probability is unchanged regardless of whether the focal individual or the opponent is the parent. Mutations occur independently in each trait and are distributed normally with the parent's trait as the mean and a

chosen standard deviation ($s_x$ or $s_n$). For a full treatment of the game details and pseudo-code, see electronic supplementary material, appendix A.

## 2.2. Estimation of branch location and probability of branching

Figure 1*b* shows examples of the stochasticity in the behaviour of the model. Six simulations are initiated with identical parameters and show considerable run-to-run variation when overlaid. Each point represents an individual, with the colour of the dot indicating the simulation. We observe that all simulations separate into two groups, but do so at different generations. We are interested in elucidating qualities of the branching probability and location in trait space as well as a quantitative description of the stochasticity of branching that we observe in figure 1*b*.

To determine the branch location in time and in trait space, we used a *k*-means clustering algorithm (as implemented in Matlab R2019b). This implementation of *k*-means clustering aims to partition *n* observations (in our case, the individuals of the population in trait space) into *k* clusters, such that each observation belongs to a cluster with the closest mean. We used *k*-means in the following way. At a branching event, the population splits into two ($k = 2$) groups. *k*-means assigns each individual to one of the two groups, and calculates the mean production of each group. A branching event was defined to occur when the means were separated by a specified threshold in their production trait. For the purposes of evaluating branching, we set this threshold to be when the means of the production trait differed by 0.2 between the two groups. Using this approach, we obtained more rigorous and quantifiable results for the probability of branching and its timing (if it occurs) for various population sizes (see electronic supplementary material, appendix B for more details).

Our two-dimensional trait space differs from the one-dimensional system in an important way; we have a finite window in trait space at which branching can occur. The population, while still monomorphic, drifts along the attracting set (solid black line in figure 2), along which branching is favoured. Once the population sufficiently decreases in neighbourhood size trait, the population moves quickly to maximum production and minimum neighbourhood size, such that branching is no longer possible. Hence, there is a finite time interval during which the population remains in a region favoured to branch. In simulations, branching or no branching occurs within a finite amount of time.

By contrast, single trait adaptive dynamics [18,39,41] observed populations that are favoured to branch for as long as the simulation runs (possibly forever). In these systems, it becomes difficult to calculate the probability that branching occurs. Our two-dimensional system does not have this problem, as we can say with certainty whether branching occurred or not within a finite time. The impact of this window will be discussed further, as it can have interesting consequences on the overall evolutionary dynamics.

# 3. Results

In this section, we assess the accuracy of the deterministic adaptive dynamics approach as it applies to our stochastic population game. First, we use the deterministic approach to make analytical predictions under the assumption of an infinitely large population. We discuss the outcome when this assumption is no longer valid, and then discuss why the approach fails and how we can adapt our predictions to capture the stochastic fluctuations that become important in small populations.

## 3.1. Monomorphic populations tend toward the branching curve

Initially, we assume a monomorphic population, with all individuals having the same production and neighbourhood size. The selection gradient, which we will denote $\vec{D}(x, n_x)$, determines the direction of evolution for population traits in the deterministic limit. To evaluate this, we first define invasion fitness, $f_{x,n_x}(y, n_y)$, which is the relative evolutionary success of a rare mutant with traits ($y, n_y$) in a resident population with traits ($x, n_x$) [18]

$$f_{x,n_x}(y, n_y) = P(y, n_y, x) - P(x, n_x, x). \tag{3.1}$$

As Dieckmann and Law showed, for a given trait, the derivative of invasion fitness with respect to the mutant's trait, evaluated at the resident trait, gives the selection gradient [15]. For a more comprehensive treatment of the monomorphic population and selection gradient, see electronic supplementary material, appendix C. In the monomorphic population, there is an absence of selective pressure on the

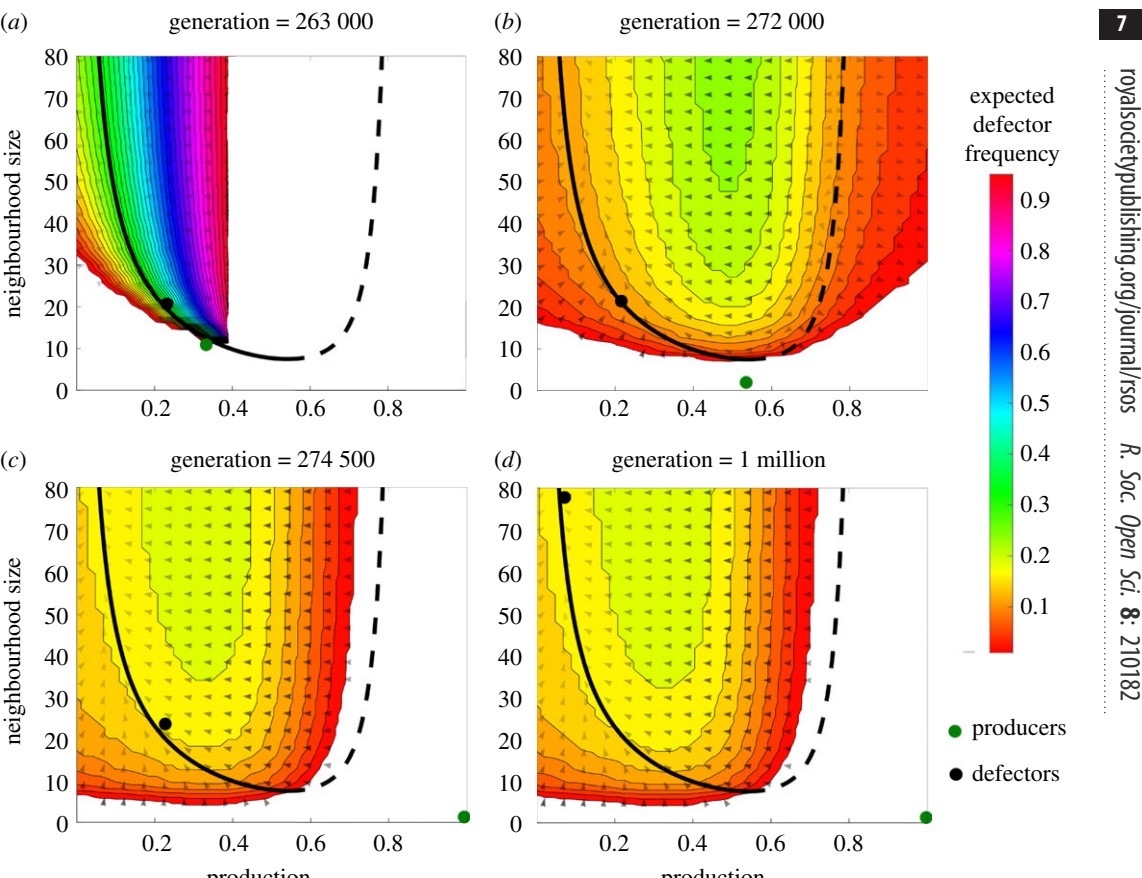

**Figure 2.** Adaptive dynamics branching prediction. The green dot represents the producer subpopulation at the given generation in a simulation of 150 individuals, while the black dot indicates the defector subpopulation. A population size of 150 was chosen because it branches consistently, but also occasionally observes extinction events following branching. Filled contour region indicates where the defectors can exist in the deterministic limit, with the colour indicating their equilibrium frequency, $p^*$. Blank region indicates where heterogeneity cannot exist, because either defectors ($p^* > 1$) or producers ($p^* < 0$) are expected to dominate based on the equilibrium frequency (see electronic supplementary material, appendix E). Arrows indicate the vector field, with opacity denoting magnitude. The solid and dashed black line indicates the equilibrium points at which the selection gradient is zero for a monomorphic population. Solid represents the attracting set, and dashed represents the repulsive set. The initial traits of the population are normally distributed with production of 0.5 and neighbourhood size of 30. 'Default' parameters ($\beta = 5$, $\sigma = 2$, $\kappa = 0.5$, $\omega = 2$) were used for this simulation. (a) Immediately following initial separation, we see the subpopulations driven further apart. There is a large span in the equilibrium frequency of the subpopulations. (b) Defector subpopulation has moved little within the weak field, but producers make their way quickly towards full production and minimum neighbourhood size. The equilibrium frequency of defectors begins to settle. (c) Producers continue moving towards full production and have roughly reached minimum neighbourhood size. (d) If we let many generations pass, we see that the defector subpopulation responds to the underlying deterministic gradient, and slowly increases its neighbourhood size accordingly and settles at low production.

neighbourhood size trait, so we have a selection gradient acting only on the production trait, $D_x(x, n_x)$

$$D_x(x, n_x) = \left.\frac{\partial f_{x,n_x}(y, n_y)}{\partial y}\right|_{y \to x, n_y \to n_x}.$$ (3.2)

For our model, the two-dimensional selection gradient is given by

$$\vec{D}(x, n_x) = (D_x, D_{n_x}) = \left(\frac{\beta\, e^{\sigma+\beta x}(1 + e^{\sigma})}{n_x(e^{\sigma} + e^{\beta x})^2} - \frac{2\kappa\omega \tanh(x/(1-x))^{\omega}}{\sinh(2x/(1-x))(x-1)^2}, 0\right).$$ (3.3)

Then, we can find the equilibrium points by finding where the selection gradient is zero. For our model,

this is a curve in trait space given by

$$n_x(x) = \frac{e^{x\beta+\sigma}(1+e^{\sigma})(x-1)^2\beta\cosh(x/(1-x))\sinh(x/(1-x))\tanh(x/(1-x))^{-\omega}}{(e^{\beta x}+e^{\sigma})^2\kappa\omega}. \qquad (3.4)$$

Parts of the curve can either be attractors of the monomorphic population, or repellors. The type can be obtained by differentiating the selection gradient with respect to $x$, as shown in electronic supplementary material, appendix C. The curve of equilibrium points is shown in figure 2. The dashed part of the curve repels the population. By contrast, the solid line shows the attracting set. Once the population approaches the solid line, it drifts along the curve and rich dynamics are possible; this is the focus of the next sections.

## 3.2. Populations along the attracting set favour deterministic branching via saddle-like instability

The attracting set is prone to invasion when the mutant fitness exceeds the resident population's fitness. In other words, in this case, the attracting set would not be a set of evolutionary stable states (ESS). Then, no longer assuming a monomorphic population, there are three possible outcomes:

1. The mutant replaces the entire resident population. At this point, the population is again monomorphic, with the 'mutant' becoming the new resident. The new resident may be perturbed slightly away from the attracting set. In that case, the perturbed monomorphic population will evolve back towards the attracting set. This may result in drift up or down along the curve of attracting points.
2. The mutant goes extinct by chance. In the strict view, where there is only resident and a mutant of greater fitness, this type of extinction would not occur in our model. However, allowing for multiple, simultaneous mutations in our model, or a different model choice altogether, this is possible. The population remains monomorphic.
3. The resident and mutant coexist, and our assumption of a monomorphic population no longer holds. Depending on the model, disruptive selection can drive the two subpopulations further apart (branching), or the subpopulations may evolve to re-combine into one monomorphic population, which may also be perturbed from the original resident trait [59].

In our simulations, we observe both branching and drift along the attracting set. We will now show that adaptive dynamics predicts branching in the case of our model. To do this, we must determine whether a point $(x, n_x)$ is a local maximum of the invasion fitness, $f_{x,n_x}(y, n_y)$. In this case, we would be at an ESS. This leads us to calculate the determinant of the Hessian of the invasion fitness function

$$\det(H(f_{x,n_x}(y, n_y))) = \left[\frac{\partial^2 f_{x,n_x}(y, n_y)}{\partial y^2}\frac{\partial^2 f_{x,n_x}(y, n_y)}{\partial n_y^2} - \left(\frac{\partial^2 f_{x,n_x}(y, n_y)}{\partial y\partial n_y}\right)^2\right]\Big|_{y\to x, n_y\to n_x} < 0. \qquad (3.5)$$

The negative determinant indicates that each point along the attracting set in trait space is a saddle point with respect to invasion fitness, and therefore unstable. This permits the possibility of branching at any point on the attracting set, because the resident is never a local maximum of the invasion fitness [60].

We can determine if the resident and mutant can coexist. An important aspect of our payoff function is frequency dependence. The payoff to one individual is dependent on the amount of production that is being generated by other individuals in their neighbourhood. If we draw neighbourhoods randomly, their composition reflects the composition of the population as a whole. Therefore, it is possible that a mutant is more fit than the resident when there is only a single mutant, but less fit once the population is composed of a sizable proportion of mutants. By finding the frequency of resident and mutant at which the payoffs are equal, which we call equilibrium frequency, we can determine if coexistence is possible. The method for evaluating the equilibrium frequency at which the payoffs are equal is discussed further in electronic supplementary material, appendix E.

We find that coexistence is favoured in the region immediately surrounding each point along the attracting set. One such example of our calculated equilibrium frequency, denoted by $p^*$, is shown as the coloured contours in figure 2, which indicate the equilibrium frequency of the defector subpopulation at various points in trait space. The coloured region indicates the defector subpopulation location at which coexistence is predicted to be favoured. This region is dependent on

the location of the producer subpopulation. For the purposes of figure 2, we set the producer subpopulation as the green dot, assigning it to four locations in trait space that were observed in a single simulation. Then, we calculate the population frequency at which the fitness of a producer would be equal to the fitness of a defector. If the fraction of defectors is between 0 and 1, then coexistence is favoured, and the contours in figure 2 show the equilibrium fraction of defectors in those regions. Note that in a finite population, the equilibrium fraction must be high enough to allow one individual (e.g. if the equilibrium fraction is 0.001 in a population of 100, coexistence would not be favoured).

Having established that coexistence is possible, branching simply requires that the selection acting on our two subpopulations drives them further apart. The negative determinant of the Hessian of invasion fitness (equation (3.5)) implies the attracting set is composed of a set of non-isolated saddle points, which is sufficient to show that disruptive selection leads to branching in the vicinity of an attracting point [60]. However, we can also show this explicitly. First, we assume a dimorphic population (mutant and resident subpopulations) and proceed in calculating the selection gradient of each subpopulation [18]. The general principles of calculating the selection gradient remain the same with two subpopulations, and the details can be found in electronic supplementary material, appendix E. Figure 2 shows the results of the calculations of the dimorphic selection gradient. The vector field shows the selection gradient acting on the defectors. Again, this requires us to set the producer subpopulation (green dot in figure 2) to a specific location in trait space, as we do for the calculation of equilibrium frequency. We see that, while selection is weak at first, the populations are driven apart and therefore branching is predicted to occur. In figure 2, we show one example, but the conditions required for branching are also satisfied at every point along the attracting set.

## 3.3. Small population size affects the probability of branching and its aftermath

Following the observations of delayed branching in figure 1b, we see even more interesting dynamics at slightly smaller population sizes. Though the predicted branching behaviour is shown for a population of 150 in figure 2, this does not always occur. Instead, figure 3 shows the full extent of the prototypical results of our simulations for a population of 200. Qualitatively distinct outcomes occur for initially identical simulations. Each point in the upper panel of figure 3 represents an individual in trait space with the colour of the dot indicating the generation. There is only one simulation shown in each panel. Each point in the lower panel of figure 3 also represents an individual, though we only show the production (vertical axis) of the individuals as it evolves in generational time steps (horizontal axis). Figure 3b,d,f (lower panel) shows the corresponding levels of production over time for each simulation shown in figure 3a,c,e (upper panel). That is, the lower panel provides a different view of the same simulation shown in the corresponding upper panel. We can classify all simulations into the three observed scenarios:

1. Sustained branching (figure 3a,b): The simulation branches into distinct subpopulations and, at the end of the simulation, there are still two subpopulations in distinct groups. The final generation groupings can be seen as the two green dots in figure 3a, one at minimum production and maximum neighbourhood size (top left), and one at maximum production and minimum neighbourhood size (bottom right).
2. Branching followed by extinction of a subpopulation (figure 3c,d): The simulation branches again but, prior to the final generation, there is extinction of one subpopulation leaving a single subpopulation (green dot) in the last generation of figure 3b. In our model, extinction only occurred in the defector subpopulation (top left group in figure 2). Also, extinction of the defector subpopulation led to an ESS of producers that could not branch again.
3. No branching (figure 3e,f): The population moves as one monomorphic group until it reaches maximum production and minimum neighbourhood size (bottom right in figure 3e), at which point it is also in an ESS and cannot branch again.

These results highlight the probabilistic nature of the branching process that is not predicted by the deterministic dynamics. In §3.2, we showed that in the deterministic limit, our population game always leads to branching (see also electronic supplementary material, appendix C for pre-branching behaviour). In small populations, stochastic dynamics clearly play a large role. Motivated by the fact that, for these conditions, we expect to see coexistence [36] (table 1), we show that three scenarios are possible. First, we observe branching and coexistence of defectors and producers (figure 3a,b). Second, we see branching followed by defector extinction (figure 3c,d). Third, we see a scenario of no branching, which then leads to producers only (figure 3e,f). Note that in this third case, defectors never arise, and thus cannot go extinct. In the following, we discuss a probabilistic approach to better understand these scenarios.

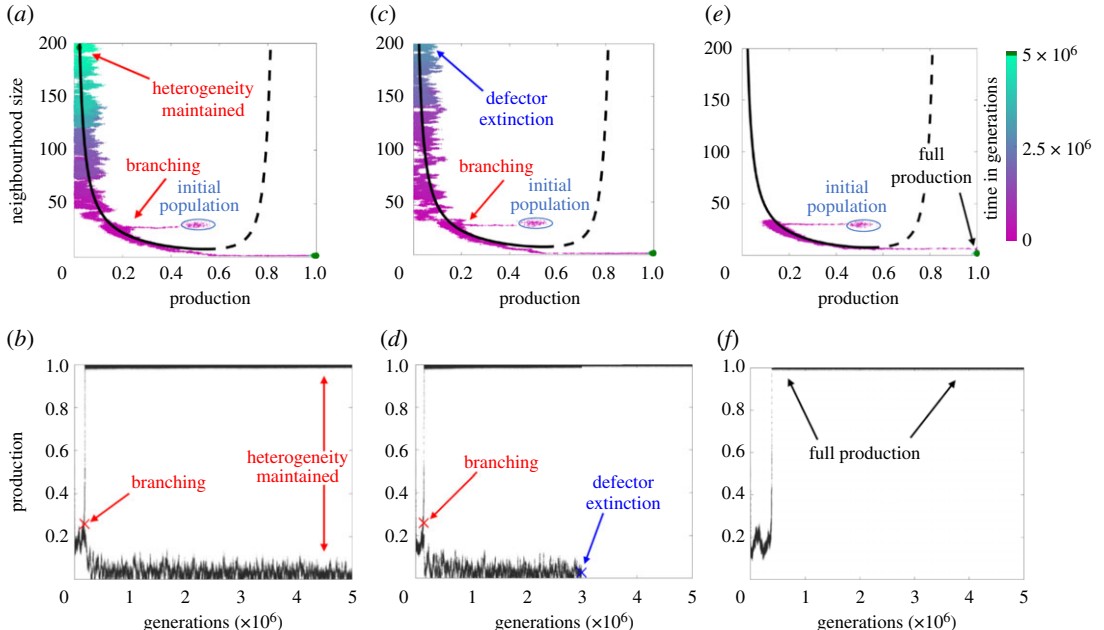

**Figure 3.** Long-term outcomes with the same population size. Each dot represents an individual. The upper panels show a trait space view, with colour indicating generation, and dark green showing the individuals of the final generation. The solid and dashed black lines show the equilibrium points at which the selection gradient is zero for a monomorphic population. Solid represents the attracting set, and dashed represents the repulsive set. The lower panels (*b*,*d*,*f*) show the corresponding production trait evolution over time. Initial states of all populations are normally distributed with production of 0.5 and neighbourhood size of 30. 'Default' parameters ($\beta = 5$, $\sigma = 2$, $\kappa = 0.5$, $\omega = 2$) were used for all simulations, and all simulations contain 200 individuals run for 5 million generations. A population size of 200 was chosen because it was the largest population to observe all three events (branching, no branching, and extinction). (*a*,*b*) Branching occurs, and is maintained, as noted by the two separate groups of dark green at the final generation. Red 'X' in *b* denotes branching point. The final result is a heterogeneous population. (*c*,*d*) Branching occurs, but extinction of the defector subpopulation occurs before the final generation, as noted by the lack of a dark green group in the upper left. Again, red 'X' in *d* denotes branching point. Additionally, blue 'X' denotes extinction point. The final result is a monomorphic population of producers, which cannot branch again. (*e*,*f*) Branching never occurs, and will never occur. The population moves out of the region where branching is possible. The result is also an all-producer population.

## 3.4. Numerical branching and extinction results

Despite the deterministic prediction that branching should occur, many of the simulated populations fail to branch. There can be qualitatively different behaviours which are inherently probabilistic, as seen in figure 3. We quantify this behaviour by running multiple simulations with identical inputs. The results give us the probability of branching at any given population size, shown in figure 4*a*. Additionally, in figure 4*b*, we find the rate of extinction for a given population size, by counting the generations from branching to extinction, if it occurs. Specifically, we begin counting for extinction when the producer subpopulation reaches the state of maximum production and minimum neighbourhood size.

When we disregard a main assumption of the adaptive dynamics approach, that of sufficiently large population size, we do not expect it to capture the whole process. Indeed, we know that a lower limit at which branching is no longer favourable exists. The trivial choice when there is only one individual cannot branch. However, these results indicate that the inability to branch occurs for populations exceeding 100 individuals.

From this analysis, we can conclude that, in small populations, a simple adaptive dynamics approach may fail. However, in line with Wakano and Iwasa, our results seem to indicate three distinct possibilities; one where branching is impossible, one where branching is stochastic, and one where branching is deterministic [39]. In the case of stochastic branching, the population may, after reaching a singular point, remain monomorphic for a period of time before branching, or, as mentioned in §2.2, may miss the opportunity to branch. Deterministic branching, on the other hand, occurs almost immediately after the population reaches a singular point.

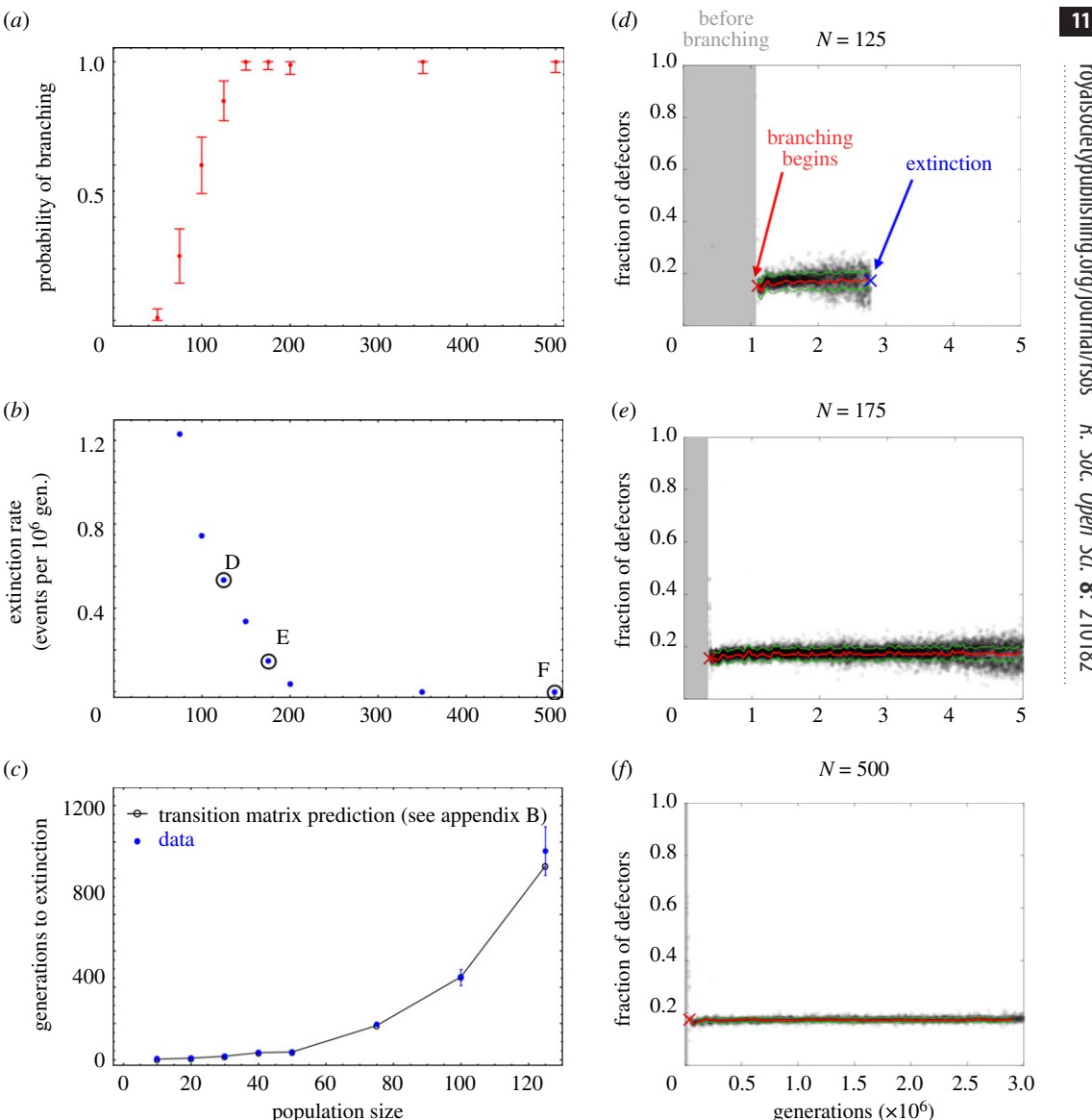

**Figure 4.** Extinction and branching events strongly depend on population size, N. One hundred simulations used for (a–b) except for N = 350 and N = 500, where only 60 were used. (a) Branching probabilities for various population sizes. The calculation of 95% confidence interval via the Agresti–Coull interval method [61]. (b) Extinction rate for various population sizes, given in the expected number of extinction events occurring in one million generations. Extinction in this case requires branching to occur first. Black circles denote example populations shown in d–f. (c) Extinction time if traits cannot mutate. In this case, we set two initial populations at minimum and maximum allowed traits (full producers and defectors), with no mutations. Note that this is a distinctly restricted model and the times to extinction are much shorter than observed in simulations with mutation, such as those shown in the other panels. Extinction occurred even if branching would not have happened first. The transition matrix approach is discussed in §3.5 and detailed further in electronic supplementary material, appendix D. Error bars for data represent the standard deviation of the mean. 300 simulations used for N = 10 − 80. One hundred simulations for N = 100 and N = 125. (d–f) Frequency of defectors for one simulation at N = 125, 175, 500, respectively. Grey shaded area indicates generations where the population was still monomorphic (e.g. before branching has occurred). Branching begins at red 'X'. Red line indicates equilibrium frequency (p*) based on traits (see electronic supplementary material, appendix E). Green indicates plus and minus one standard deviation based on simulated data. Extinction occurs at blue 'X' only for d.

## 3.5. The cause of defector extinction

Numerical results suggest that extinction and branching are related. Therefore, we sought to elucidate this relationship in greater detail. We identified two possibilities, either extinction occurs because mutational drift moves a subpopulation into a region of trait space unfavourable for coexistence, or

deviations in subpopulation frequency from the mean lead to reaching an absorbing state. The latter can occur despite the fact that the subpopulation is in a favourable region for coexistence. While adaptive dynamics may not perfectly capture the extinction and branching possibilities, it still makes reasonable predictions on the mean frequencies of the subpopulations. We use the data from our $k$-means analysis, and expand upon the discussion of subpopulation frequency in Doebeli *et al.* to find an expected frequency which will allow us to determine whether a subpopulation was located in a 'favourable region for coexistence' [18] (see electronic supplementary material, appendix C). This approach will then allow us to make predictions about extinction, as shown in figure 4c and detailed further below.

Given $p^*$, we can determine the primary cause of extinction. The expected frequency of defectors is shown as the red line in figure 4d–f. The black dots in figures 4d–f show the observed frequency over time in the simulation. In smaller populations, we see that the variance in the fraction of defectors is larger, but is nonetheless centred on the predicted frequency. One of the two previously discussed possibilities for extinction is that the fitness of defectors is too low to sustain coexistence. This would be indicated by the red line going to zero in figure 4d–f. Clearly, this does not occur because the expected defector frequency is relatively stable (red line), showing only minor fluctuations at the beginning and eventually levelling off to a constant value near $p^* = 0.17$. Therefore, extinction must occur due to stochastic fluctuations in subpopulation frequency causing it to abruptly hit the absorbing state, which in this case is a state of all producers. We observe this frequency-based extinction in figure 4d, where the expected frequency is stable but extinction still occurs. This extinction can have significant impacts on the overall evolution of the system, as is the case for our model. As shown in figure 4, extinction leads to an all producer state, which is an evolutionary stable strategy and is not viable to branch again.

This finding allows us to predict the time to extinction in certain cases using a transition matrix approach, the results of which are shown in figure 4c. We employ a few simplifying assumptions, removing mutations and seeding an already dimorphic population, with one subpopulation at maximum production and minimum neighbourhood size (producers) and one subpopulation at minimum production and maximum neighbourhood size (defectors). Within each subpopulation, the individuals share identical traits. Therefore, a producer selecting a producer as its opponent does not change the composition of the population; either way a producer will be the offspring. In the same way, a defector with an opponent who is also a defector does not change the population. However, in the case where a defector (producer) is the focal individual and the producer (defector) is the opponent, we have some probability of losing a defector (producer). The absorbing states of these transitions are those in which our population once again becomes monomorphic (all producers or all defectors). In this simplified case of our model, if defectors (producers) are overabundant, any focal individual who is a defector (producer) will be replaced by a producer (defector), if a producer (defector) is their randomly assigned opponent.

Knowing the number of producers and defectors at any given generation, we can calculate which subpopulation is overabundant. For example, if defectors are overabundant, we focus on the opponents that each one of them selects. For each defector which selects an opponent that is a producer (which in this case would be a more fit opponent), the defector subpopulation decreases by one. The probability of a defector choosing a producer as its opponent is dictated by the composition of the population. Calculating the probability of moving from a current state of $m$ defectors to a state of $n$ defectors in the next generation, we can generate probabilities for movement between states in the matrix, where rows indicate the current number of defectors, and columns indicate the number of defectors in the next generation. The same can be done when the producers are overabundant, making defectors more fit. Combining the two cases (defectors overabundant and producers overabundant), we generate a full matrix which, for each row $m$, indicates the probability of moving to a state of $n$ defectors in the next generation. The full details are shown in electronic supplementary material, appendix D.

# 4. Discussion

Our results highlight the utility and shortcomings of deterministic adaptive dynamics approaches to explain stochastic population dynamics. First, we demonstrated that the adaptive dynamics approach used to predict branching in one-dimensional games can be extended to our game with two coevolving traits. Figure 2 shows the predicted post-branching dynamics which align very well with

the simulation results in which branching occurred but not extinction. Then, we showed that, depending on population size, there is a threshold below which deterministic approaches largely fail to capture the dynamics. Figure 3 demonstrates the possible outcomes, including two alternative outcomes, extinction and no branching, which are not captured by deterministic theory. We then elucidated the relationships between extinction events by determining that frequency fluctuation was the primary cause of extinction, as shown in figure 4d. A simplified model was used to predict extinction using a transition matrix approach, as shown in figure 4c, with good agreement.

The threshold at which adaptive dynamics no longer captures the behaviour is probably trait- and game-dependent and drives the emergence of important phenomena such as probability and timing of branching or extinction events. We here examined the extent to which this stochasticity impacts evolution, and determined the population size that is large enough to avoid the impact of extinction. In future work, such a modelling approach can be used to quantify the effects of the PGs game's key parameters on branching and extinction.

Up to now, analytical approaches to capture branching in finite populations were only applied to single trait evolution. Wakano and Iwasa show that there are three possible evolutionary scenarios [39], which is consistent with our results, matching closely with the branching probabilities we report in figure 4a, despite the difference in dimensionality. Débarre & Otto [41] came to similar, single-trait conclusions even after modifying the evolutionary game to allow for a varying population size. Although we here kept population size fixed, we extend the adaptive evolutionary game dynamics to a two-dimensional trait space, which led to two important findings. First, our simulations support the idea that the theory of finite population branching can be generalized to more than one dimension. Second, we show that there exists a finite window in time in which branching can occur, leading to possibly much faster simulation approaches. If the stochastic process does not branch by the time the population moves out of the specified window in which branching is favoured, it will never branch.

The appearance of a finite branching time window can have important consequences for real evolving populations, which may present with small population sizes and multiple co-dependent traits. If the 'window' is missed, the course of evolution in such populations could be changed irreversibly. For this reason, it would be useful to further extend and analyse the work of Wakano & Iwasa to multiple dimensions [39].

Our model also has some potential limitations. We model selection as a pairwise comparison between a focal individual and a randomly chosen opponent. A continuous-time birth–death process that attempts to more closely mirror cell replication may shift the results. In addition, the extinction we observe is often a result of the discrete generational time step. If defectors automatically became more fit as soon as they were less frequent, extinction might be less likely. A comparison between these two implementations would be helpful to further our understanding of the impact of small populations.

It would be useful to know if small population effects are relevant in larger populations. The construction of the interaction neighbourhood could play an important role and was not considered in detail here. We assumed that a focal cell randomly chooses its neighbours, but perhaps it can change its interaction radius as it searches locally. That is, suppose an individual selects neighbours within a fixed radius. Based on the composition it observes in its neighbourhood it may choose to increase or decrease that radius in the future. This additional adaptive potential would impose a type of directed motion in trait space.

There are also many variations on our game which might produce additional interesting insights. While we implemented a cost in PG production, there is also the possibility of implementing a cost to neighbourhood size. Such a cost could, for example, be thought of as a natural drawback to increased motility or chemical sensing as the cell searches for more favourable environments. Similarly, it would be interesting to implement an adaptation of our game within the framework of a fixed benefit and shared cost, leading to a multi-player 'snowdrift' game.

We decided to model a PG which can be used by multiple individuals without being exhausted/consumed. This is a specific case which would not be directly applicable to a PG that must be consumed and/or depleted. Therefore, our model could produce new and interesting results if we extend it to incorporate such a scenario where the PG is consumed by the individuals. This would be likely to produce very different behaviour as a result of the fact that it would be more difficult for defectors to access the 'private' good of producers. The specific model choice would be important in this case. For example, would a neighbourhood size of 1 mean that all of the good produced by an individual is consumed by that individual, or would the neighbourhood still be invaded by defectors? Of course, the best fit for any model will depend on the exact situation which is being studied but, with some tweaking, the general framework can be applicable to a variety of real-life cases.

As we have shown, extinction often occurs due to the stochastic fluctuations about the theoretical equilibrium frequency, $p^*$. We also note that this process's variance drops off as $1/\sqrt{N}$ for larger populations. However, smaller populations seemed to reach an upper limit in their deviations, possibly due to relevant covariances or due to the selective bias of extinction, as exceeding a certain deviation would cause extinction bias. We also notice a possible link between extinction and branching. Where branching becomes much more likely, the extinction rate drops quickly to near zero. Perhaps the process that limits branching in the first place is similar to, or the same as, the process which causes extinction. Thus, our work serves to highlight the richness of adaptive dynamics models in the stochastic regime.

Data accessibility. Data and relevant code for this research work are stored in GitHub: https://github.com/MathOnco/adaptivePGG and have been archived within the Zenodo repository: https://doi.org/10.5281/zenodo.4672572.

Competing interests. We declare we have no competing interests.

Funding. This research was supported by the National Cancer Institute, part of the National Institutes of Health, under grant no. P30-CA076292. The content is solely the responsibility of the authors and does not necessarily represent the official views of the National Institutes of Health or the H. Lee Moffitt Cancer Center and Research Institute. P.M.A. acknowledges funding from Florida Health's Bankhead Coley grant no. 20B06, and from USAMRAA grant no. KC180036. The authors acknowledge the financial support of the Frank E. Duckwall Foundation and the Richard O. Jacobson Foundation.

Acknowledgements. The authors thank the Moffitt Summer Undergraduate Program to Advance Research Knowledge (SPARK) for providing a platform that led to early results presented in this article.

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
