## [Peer Review File · Royal Society Open Science]

Review History

RSOS-210182.R0 (Original submission)

Review form: Reviewer 1

Is the manuscript scientifically sound in its present form?

Yes

Are the interpretations and conclusions justified by the results?

Yes

Is the language acceptable?

Yes

Do you have any ethical concerns with this paper?

Yes

Have you any concerns about statistical analyses in this paper?

No

Recommendation?

Accept with minor revision (please list in comments)

Comments to the Author(s)

In this manuscript, the authors study the adaptive dynamics of production and neighborhood size in the nonlinear public goods game where the benefit and cost functions are nonlinear when the population size is large. By performing theoretical analysis and numerical simulation, they reveal that the population size plays an important role in determining the final state of the population. Furthermore, when the population size is small, the authors calculate the probability of extinction of a subpopulations and capture some interesting evolutionary outcomes which are not captured by deterministic theory. I find that the work is of broad interest. However, I still have some following comments or questions on it.

(1) In page 1, the authors use the abbreviation PG in the abstract, but does not give the full name.

(2) In page 3, what does 1D games mean? The authors should give an explanation.

(3) In page 4, the authors state that in the evolutionary process the neighborhood size-trait is also a continuous variable between 1 and N. But the authors also need to consider that the neighborhood size should be an integer.

(4) I do not see the parameter values which are used to plot figure 2 and figure S1.

(5) In page 7, the probability that the opponent replaces the individual and becomes the parent of an offspring in the next generation (q) will be negative when $P_{opp} < P_{focal}$. The authors should clarify this.

(6) The public goods dilemma frequently appears in the real world, and some works about the public goods game can be mentioned the literature review, e.g., *Journal of the Royal Society Interface*, 2013, 10, 20120997; *New Journal of Physics*, 2014, 16: 083016; *Mathematical Models and Methods in Applied Sciences*, 2019, 29: 2127--2149; *Nonlinear Dynamics*, 2019, 97:749-766.

(7) There are some typos in the work. For example, in page 17, line 29-32, "First, we demonstrated that Then, we show that..."; In page 13 of SI, Refs [1-4] are missing; In page 13 of SI, line 3, "...know what what direction...". The authors should proof-read again and have the spelling and language checked.

Review form: Reviewer 2

Is the manuscript scientifically sound in its present form?

Yes

Are the interpretations and conclusions justified by the results?

Yes

Is the language acceptable?

Yes

Do you have any ethical concerns with this paper?

No

Have you any concerns about statistical analyses in this paper?

No

Recommendation?

Accept with minor revision (please list in comments)

Comments to the Author(s)

This paper studies the adaptive dynamics of nonlinear public goods game in a population with evolving neighborhood size. The authors find the spontaneous emergence of two distinct subpopulations comprised of producers who make large investments and free-riders who contribute very little, each with a different neighborhood size from an initially monomorphic population in trait space driven by evolutionary branching. Furthermore, the intrinsic stochasticity arising from population size is shown to play a dominant role in determining the final state of the population, i.e., branching or extinction.

I think the novelty of the work lies in its consideration of the co-evolution of neighborhood size and public good production, which is, as far as I know, not investigated in previous papers focusing on adaptive dynamics. Besides, the theoretical analysis of adaptive dynamics employed seems to be also sound. Considering the acceptance criterion of Royal Society Open Science, I support its publication if the following suggestions are incorporated.

(1) The evolutionary branching of cooperators and defectors in social dilemma games is not new. In 2004, Doebeli et al. discover the evolutionary origin of cooperators and defectors in the continuous snowdrift game governed by pairwise interactions [The Evolutionary Origin of Cooperators and Defectors. *Science* 306, 859-862 (2004)]. A more relevant and recent work has been done by Zhang et al. in 2013, who have studied the adaptive dynamics of nonlinear public goods governed by collective interactions in finitely well-mixed populations [A tale of two contribution mechanisms for nonlinear public goods. *Sci. Rep.* 3, 2021 (2013)]. They have shown that nonlinearity of public goods production alone is sufficient for inducing the phenomenon of evolutionary branching. Then what particular role the evolving population structure, i.e., the evolving neighborhood size, plays in the adaptive dynamics of nonlinear public goods? Considering above facts, I suggest the authors to compare the results with that in [A tale of two contribution mechanisms for nonlinear public goods. *Sci. Rep.* 3, 2021 (2013)].

(2) An important issue I do feel confused is the way in which the evolving population structure, or more specifically, the evolving neighborhood size, is modelled. I would prefer if the authors can give more specific examples that indeed support such a modelling manner of population structure.

(3) With reference to the nonlinear relationship between the common benefits and the overall contributions, I suggest citing the following very relevant paper [Impact of critical mass on the evolution of cooperation in spatial public goods games. *Phys. Rev. E* 81, 057101 (2010)]. Besides, regarding the multi-player social dilemma games on structure populations, I recommend the author to cite a recent review paper [Statistical physics of human cooperation. *Phys. Rep.* 687, 1-51 (2017)] to strengthen the general background.

(4) There are a few English mistakes or vague expressions in the manuscript. Please carefully proofread the article before resubmission.

Review form: Reviewer 3**Is the manuscript scientifically sound in its present form?**

Yes

Are the interpretations and conclusions justified by the results?

Yes

Is the language acceptable?

Yes

Do you have any ethical concerns with this paper?

No

Have you any concerns about statistical analyses in this paper?

Yes

Recommendation?

Accept with minor revision (please list in comments)

Comments to the Author(s)

The tragedy of commons is a long-standing puzzle, and public goods game (PGG) theory provides a potential route to resolve it. In this paper, the authors explored the effect of nonlinear benefit and cost function on the branching and extinction in PGG, and obtained that population size would be a crucial role in determining the final state of the population.

In my opinion, the current idea is interesting, and I am willing to consider the potential recommendation. Yet, before the formal acceptance, I have some technical comments on the related contents as follows:

1. As the authors state, in Figure 1 B, the branch point location and probability in trait space are interesting. However, I was wondering whether the qualities of the branch point location and probability can be shown more obviously in Fig 1 B, or can be marked as some more detailed number of generations.
2. Population's size plays a crucial role in determining the final state. Then, how to ascertain the population's size needs to be described. Meanwhile, is the assumption of an infinitely large population reasonable in practice and why the small population number is selected as 150, 200?
3. The English spelling need to be checked carefully. For example, P6: Line 40, "...shown for both the monorphic...." should revised as "...shown for both the monomorphic...". P12 Lines 14, 32, etc "...figure ..." should be "...Figure 2...".
4. Regarding the PGG, many mechanisms have been proposed to enrich the understanding of the emergence of cooperation. As an example, the adaptive reputation mechanism is an effective one, two recent works [Evolution of cooperation in the spatial public goods game with adaptive reputation assortment. *Physics Letters A*. 2016,380: 40-47; Effect of memory, intolerance and second-order reputation on cooperation. *Chaos*, 2020, 30: .063122.] are worth mentioning here.
5. The references in Appendix D are empty !

Decision letter (RSOS-210182.R0)

Dear Dr Kimmel

On behalf of the Editors, we are pleased to inform you that your Manuscript RSOS-210182 "Branching and extinction in evolutionary public goods games" has been accepted for publication

in Royal Society Open Science subject to minor revision in accordance with the referees' reports. Please find the referees' comments along with any feedback from the Editors below my signature.

Please submit your revised manuscript and required files (see below) no later than 7 days from today's (ie 31-Mar-2021) date. Note: the ScholarOne system will 'lock' if submission of the revision is attempted 7 or more days after the deadline. If you do not think you will be able to meet this deadline please contact the editorial office immediately.

on behalf of Professor Matjaz Perc (Associate Editor) and Mark Chaplain (Subject Editor)
openscience@royalsociety.org

Reviewer comments to Author:
Reviewer: 1
Comments to the Author(s)

In this manuscript, the authors study the adaptive dynamics of production and neighborhood size in the nonlinear public goods game where the benefit and cost functions are nonlinear when the population size is large. By performing theoretical analysis and numerical simulation, they reveal that the population size plays an important role in determining the final state of the population. Furthermore, when the population size is small, the authors calculate the probability of extinction of a subpopulations and capture some interesting evolutionary outcomes which are not captured by deterministic theory. I find that the work is of broad interest. However, I still have some following comments or questions on it.

- (1) In page 1, the authors use the abbreviation PG in the abstract, but does not give the full name.
- (2) In page 3, what does 1D games mean? The authors should give an explanation.
- (3) In page 4, the authors state that in the evolutionary process the neighborhood size-trait is also a continuous variable between 1 and N. But the authors also need to consider that the neighborhood size should be an integer.

(4) I do not see the parameter values which are used to plot figure 2 and figure S1.

(5) In page 7, the probability that the opponent replaces the individual and becomes the parent of an offspring in the next generation (q) will be negative when P_{opp}

(6) The public goods dilemma frequently appears in the real world, and some works about the public goods game can be mentioned the literature review, e.g., *Journal of the Royal Society Interface*, 2013, 10, 20120997; *New Journal of Physics*, 2014, 16: 083016; *Mathematical Models and Methods in Applied Sciences*, 2019, 29: 2127--2149; *Nonlinear Dynamics*, 2019, 97:749-766.

(7) There are some typos in the work. For example, in page 17, line 29-32, "First, we demonstrated that Then, we show that..."; In page 13 of SI, Refs [1-4] are missing; In page 13 of SI, line 3, "...know what what direction...". The authors should proof-read again and have the spelling and language checked.

Reviewer: 2

Comments to the Author(s)

This paper studies the adaptive dynamics of nonlinear public goods game in a population with evolving neighborhood size. The authors find the spontaneous emergence of two distinct subpopulations comprised of producers who make large investments and free-riders who contribute very little, each with a different neighborhood size from an initially monomorphic population in trait space driven by evolutionary branching. Furthermore, the intrinsic stochasticity arising from population size is shown to play a dominant role in determining the final state of the population, i.e., branching or extinction.

I think the novelty of the work lies in its consideration of the co-evolution of neighborhood size and public good production, which is, as far as I know, not investigated in previous papers focusing on adaptive dynamics. Besides, the theoretical analysis of adaptive dynamics employed seems to be also sound. Considering the acceptance criterion of Royal Society Open Science, I support its publication if the following suggestions are incorporated.

(1) The evolutionary branching of cooperators and defectors in social dilemma games is not new. In 2004, Doebeli et al. discover the evolutionary origin of cooperators and defectors in the continuous snowdrift game governed by pairwise interactions [The Evolutionary Origin of Cooperators and Defectors. *Science* 306, 859-862 (2004)]. A more relevant and recent work has been done by Zhang et al. in 2013, who have studied the adaptive dynamics of nonlinear public goods governed by collective interactions in finitely well-mixed populations [A tale of two contribution mechanisms for nonlinear public goods. *Sci. Rep.* 3, 2021 (2013)]. They have shown that nonlinearity of public goods production alone is sufficient for inducing the phenomenon of evolutionary branching. Then what particular role the evolving population structure, i.e., the evolving neighborhood size, plays in the adaptive dynamics of nonlinear public goods? Considering above facts, I suggest the authors to compare the results with that in [A tale of two contribution mechanisms for nonlinear public goods. *Sci. Rep.* 3, 2021 (2013)].

(2) An important issue I do feel confused is the way in which the evolving population structure, or more specifically, the evolving neighborhood size, is modelled. I would prefer if the authors can give more specific examples that indeed support such a modelling manner of population structure.

(3) With reference to the nonlinear relationship between the common benefits and the overall contributions, I suggest citing the following very relevant paper [Impact of critical mass on the evolution of cooperation in spatial public goods games. *Phys. Rev. E* 81, 057101 (2010)]. Besides, regarding the multi-player social dilemma games on structure populations, I recommend the author to cite a recent review paper [Statistical physics of human cooperation. *Phys. Rep.* 687, 1-51 (2017)] to strengthen the general background.

(4) There are a few English mistakes or vague expressions in the manuscript. Please carefully proofread the article before resubmission.

Reviewer: 3
 Comments to the Author(s)

The tragedy of commons is a long-standing puzzle, and public goods game (PGG) theory provides a potential route to resolve it. In this paper, the authors explored the effect of nonlinear benefit and cost function on the branching and extinction in PGG, and obtained that population size would be a crucial role in determining the final state of the population.

In my opinion, the current idea is interesting, and I am willing to consider the potential recommendation. Yet, before the formal acceptance, I have some technical comments on the related contents as follows:

1. As the authors state, in Figure 1 B, the branch point location and probability in trait space are interesting. However, I was wondering whether the qualities of the branch point location and probability can be shown more obviously in Fig 1 B, or can be marked as some more detailed number of generations.
2. Population's size plays a crucial role in determining the final state. Then, how to ascertain the population's size needs to be described. Meanwhile, is the assumption of an infinitely large population reasonable in practice and why the small population number is selected as 150, 200?
3. The English spelling need to be checked carefully. For example, P6: Line 40, "...shown for both the monorphic...." should revised as "...shown for both the monomorphic...". P12 Lines 14, 32, etc "...figure ..." should be "...Figure 2...".
4. Regarding the PGG, many mechanisms have been proposed to enrich the understanding of the emergence of cooperation. As an example, the adaptive reputation mechanism is an effective one, two recent works [Evolution of cooperation in the spatial public goods game with adaptive reputation assortment. *Physics Letters A*. 2016,380: 40-47; Effect of memory, intolerance and second-order reputation on cooperation. *Chaos*, 2020, 30: .063122.] are worth mentioning here.
5. The references in Appendix D are empty !

===PREPARING YOUR MANUSCRIPT===

===PREPARING YOUR REVISION IN SCHOLARONE===

Author's Response to Decision Letter for (RSOS-210182.R0)

See Appendix A.

RSOS-210182.R1 (Revision)

Review form: Reviewer 1

Is the manuscript scientifically sound in its present form?

Yes

Are the interpretations and conclusions justified by the results?

Yes

Is the language acceptable?

Yes

Do you have any ethical concerns with this paper?

No

Have you any concerns about statistical analyses in this paper?

No

Recommendation?

Accept as is

Comments to the Author(s)

The revised MS seems acceptable now.

Review form: Reviewer 3

Is the manuscript scientifically sound in its present form?

Yes

Are the interpretations and conclusions justified by the results?

Yes

Is the language acceptable?

Yes

Do you have any ethical concerns with this paper?

No

Have you any concerns about statistical analyses in this paper?

No

Recommendation?

Accept as is

Comments to the Author(s)

The authors have made the careful revision, I am willing to recommend it to be accepted in the present form.

Decision letter (RSOS-210182.R1)

Dear Dr Kimmel,

It is a pleasure to accept your manuscript entitled "Two-dimensional adaptive dynamics of evolutionary public goods games: finite size effects on fixation probability and branching time" in its current form for publication in Royal Society Open Science. The comments of the reviewer(s) who reviewed your manuscript are included at the foot of this letter.

You can expect to receive a proof of your article in the near future. Please contact the editorial office (openscience@royalsociety.org) and the production office (openscience_proofs@royalsociety.org) to let us know if you are likely to be away from e-mail contact – if you are going to be away, please nominate a co-author (if available) to manage the proofing process, and ensure they are copied into your email to the journal.

Please see the Royal Society Publishing guidance on how you may share your accepted author manuscript at <https://royalsociety.org/journals/ethics-policies/media-embargo/>. After publication, some additional ways to effectively promote your article can also be found here

<https://royalsociety.org/blog/2020/07/promoting-your-latest-paper-and-tracking-your-results/>.

on behalf of Prof Mark Chaplain (Subject Editor)
openscience@royalsociety.org

Reviewer comments to Author:
Reviewer: 3

Comments to the Author(s)
The authors have made the careful revision, I am willing to recommend it to be accepted in the present form.

Reviewer: 1

Comments to the Author(s)
The revised MS seems acceptable now.

Appendix A

Andrew Dunn, PhD
Journal editor
Royal Society Open Science

April 3, 2021

Re: Decision on Manuscript, ID: RSOS-210182

Dear Dr. Dunn,

We are delighted to learn that our manuscript entitled “Branching and extinction in evolutionary public goods games” has been accepted for publication in Open science. One of the referees has brought up a few minor comments (in *italics*). These can be seen in the “tracked changes” document. Of note, we changed the title of our manuscript to “Two-dimensional adaptive dynamics of evolutionary public goods games: finite size effects on fixation probability and branching time”.

In the attached pages, we go through the final reviewers’ comments point-by-point. We also attached a pdf highlighting all changes we made in the manuscript.

Sincerely,

Dr. Gregory Kimmel

Dr. Philipp Altrock

In the following, we address each of the referee's comments in detail, also describing what we changed in the manuscript. The referees' comments are written in black, our responses in blue.

Reviewer(s)' Comments to Author:

Reviewer: 1

Comments to the Author(s)

(1) In page 1, the authors use the abbreviation PG in the abstract, but does not give the full name.

We have changed the sentence "We allow the PG production..." -> "We allow the public good production..."

(2) In page 3, what does 1D games mean? The authors should give an explanation.

By 1D games, we mean adaptive dynamics where only a single trait is allowed to evolve over time. In this 1D setting, branching, if it should occur, always does. We have added clarification, "we observe that branching may be deterministically favored only for a finite amount of time in our two-dimensional model: the monomorphic population can drift away from the region where branching is deterministically favored, a feature not seen in the single trait games."

(3) In page 4, the authors state that in the evolutionary process the neighborhood size-trait is also a continuous variable between 1 and N. But the authors also need to consider that the neighborhood size should be an integer.

We appreciate this feedback. When determining the payoff (two paragraphs later), we describe the procedure where we round to the nearest integer when determining the neighborhood size. We have made a brief comment at the definition that this is done.

(4) I do not see the parameter values which are used to plot figure 2 and figure S1.

We have added these parameter values to the captions of figures 2 and S1, they are the values presented in the table.

(5) In page 7, the probability that the opponent replaces the individual and becomes the parent of an offspring in the next generation (q) will be negative when P_{opp}

We thank the reviewer for catching this. The simulations are all correct, we are missing that if P_{opp} is smaller, then $q = 0$. We have updated equation (6).

(6) The public goods dilemma frequently appears in the real world, and some works about the public goods game can be mentioned the literature review, e.g., Journal of the Royal Society Interface, 2013, 10, 20120997; New Journal of Physics, 2014, 16: 083016; Mathematical Models and Methods in Applied Sciences, 2019, 29: 2127--2149; Nonlinear Dynamics, 2019, 97:749-766.

We thank the reviewer for these references and have added them along with a brief description.

(7) There are some typos in the work. For example, in page 17, line 29-32, “First, we demonstrated that Then, we show that...”; In page 13 of SI, Refs [1-4] are missing; In page 13 of SI, line 3, “...know what what direction...”. The authors should proof-read again and have the spelling and language checked.

Broken refs and typos have been fixed, thank you.

Reviewer: 2

Comments to the Author(s)

(1) The evolutionary branching of cooperators and defectors in social dilemma games is not new. In 2004, Doebeli et al. discover the evolutionary origin of cooperators and defectors in the continuous snowdrift game governed by pairwise interactions [The Evolutionary Origin of Cooperators and Defectors. *Science* 306, 859-862 (2004)]. A more relevant and recent work has been done by Zhang et al. in 2013, who have studied the adaptive dynamics of nonlinear public goods governed by collective interactions in finitely well-mixed populations [A tale of two contribution mechanisms for nonlinear public goods. *Sci. Rep.* 3, 2021 (2013)]. They have shown that nonlinearity of public goods production alone is sufficient for inducing the phenomenon of evolutionary branching. Then what particular role the evolving population structure, i.e., the evolving neighborhood size, plays in the adaptive dynamics of nonlinear public goods? Considering above facts, I suggest the authors to compare the results with that in [A tale of two contribution mechanisms for nonlinear public goods. *Sci. Rep.* 3, 2021 (2013)].

We have added this work and a compare and contrast of their results with ours. “Nonlinear functions with continuous traits have been shown to be capable of branching with a single trait [Zhang, Doebeli]. In Zhang et al., the exploration of continuous investment vs. probabilistic binary investment in finite populations shows that only continuous investment allows for branching in a single trait [Zhang]. We further explore continuous investment as it relates to a new model in which there are two co-evolving traits. Analyzing two traits in our model allows us to explore the impact of finite population size in a game which features co-evolution, which is likely present in most real-world applications.”

(2) An important issue I do feel confused is the way in which the evolving population structure, or more specifically, the evolving neighborhood size, is modelled. I would prefer if the authors can give more specific examples that indeed support such a modelling manner of population structure.

We appreciate the input and have added an example which justifies our modeling of the population structure and neighborhood size.

(3) With reference to the nonlinear relationship between the common benefits and the overall contributions, I suggest citing the following very relevant paper [Impact of critical mass on the evolution of cooperation in spatial public goods games. Phys. Rev. E 81, 057101 (2010)]. Besides, regarding the multi-player social dilemma games on structure populations, I recommend the author to cite a recent review paper [Statistical physics of human cooperation. Phys. Rep. 687, 1-51 (2017)] to strengthen the general background.

We appreciate the additional recommendations and will add the first of these to our model introduction. However, we were previously advised by a reviewer to avoid human behavioral applications and for that reason we took the liberty to not add the second paper.

(4) There are a few English mistakes or vague expressions in the manuscript. Please carefully proofread the article before resubmission.

Typos have been addressed and clarity improved, thank you.

Reviewer: 3

Comments to the Author(s)

1. As the authors state, in Figure 1 B, the branch point location and probability in trait space are interesting. However, I was wondering whether the qualities of the branch point location and probability can be shown more obviously in Fig 1 B, or can be marked as some more detailed number of generations.

With several simulations overlapping, it is difficult to show exactly where each one of them branches without cluttering the figure. Instead, Fig 1 B provides a qualitative assessment of the stochasticity, which is further elucidated and quantified in Figures 2-4. We have modified the caption for figure 1 to address this. "However, the generation at which branching occurs varies, highlighting the importance of stochasticity in our game. This stochasticity will be the focus of the following analysis, as shown in greater depth in Figures 2-4."

It is difficult to define when a successful branch has occurred. We have used a K-means (with $K = 2$) detection algorithm and have stated a branching event occurs when the distance between the means is above some predefined threshold. We have marked these locations by color-coded arrows on figure 1B.

2. Population's size plays a crucial role in determining the final state. Then, how to ascertain the population's size needs to be described. Meanwhile, is the assumption of an infinitely large population reasonable in practice and why the small population number is selected as 150, 200?

The final state, probability of branching and other parts of the dynamics are all population-size dependent. We are confused by what the reviewer means with "ascertain the population size", this is a predefined quantity that remains fixed throughout the simulation. The impact of population size was investigated both numerically and analytically. The sizes of 150-200 were

chosen because these were the population sizes which branch somewhat consistently, but also observe extinction events, as seen in figure 4. For these reasons, they were ideal population sizes for further study as they are capable of all three interesting events (branching, no branching, and extinction). This explanation has been added to the caption for Figure 3.

3. The English spelling need to be checked carefully. For example, P6: Line 40, "...shown for both the monorphic....." should revised as "...shown for both the monomorphic...". P12 Lines 14, 32, etc "...figure ..." should be "...Figure 2...".

Typos have been addressed and clarity improved, thank you.

4. Regarding the PGG, many mechanisms have been proposed to enrich the understanding of the emergence of cooperation. As an example, the adaptive reputation mechanism is an effective one, two recent works [Evolution of cooperation in the spatial public goods game with adaptive reputation assortment. *Physics Letters A*. 2016,380: 40-47; Effect of memory, intolerance and second-order reputation on cooperation. *Chaos*, 2020, 30: .063122.] are worth mentioning here.

We have added these to the introduction, thank you.

5. The references in Appendix D are empty!

Fixed, thanks.